# Photodynamic Therapy with Zinc Phthalocyanine Inhibits the Stemness and Development of Colorectal Cancer: Time to Overcome the Challenging Barriers?

**DOI:** 10.3390/molecules26226877

**Published:** 2021-11-15

**Authors:** Mahsa Gholizadeh, Mohammad Amin Doustvandi, Fateme Mohammadnejad, Mahdi Abdoli Shadbad, Habib Tajalli, Oronzo Brunetti, Antonella Argentiero, Nicola Silvestris, Behzad Baradaran

**Affiliations:** 1Immunology Research Center, Tabriz University of Medical Sciences, Tabriz 51666-14731, Iran; Gmahsa42@gmail.com (M.G.); m.amin.dostvandi@gmail.com (M.A.D.); fatememohamadnejad@gmail.com (F.M.); abdoli.med99@gmail.com (M.A.S.); 2Student Research Committee, Tabriz University of Medical Sciences, Tabriz 51666-14731, Iran; 3Biophotonic Research Center, Islamic Azad University, Tabriz Branch, Tabriz 51579-44533, Iran; habibtajalli@yahoo.com; 4Research Institute for Applied Physics and Astronomy, University of Tabriz, Tabriz 51666-16471, Iran; 5Istituto Tumori BariGiovanni Paolo II, Istituto Nazionale dei Tumori (IRCCS), 70124 Bari, Italy; dr.oronzo.brunetti1983@gmail.com (O.B.); argentieroantonella@gmail.com (A.A.); 6Department of Biomedical Sciences and Human Oncology, School of Medicine, University of Bari Aldo Moro, 70124 Bari, Italy; 7Neuroscience Research Center, Tabriz University of Medical Sciences, Tabriz 51666-14731, Iran

**Keywords:** photodynamic therapy, zinc phthalocyanine, cancer stem cell, colorectal cancer, stemness, prognosis, metastasis

## Abstract

Photodynamic therapy (PDT) is a light-based cancer therapy approach that has shown promising results in treating various malignancies. Growing evidence indicates that cancer stem cells (CSCs) are implicated in tumor recurrence, metastasis, and cancer therapy resistance in colorectal cancer (CRC); thus, targeting these cells can ameliorate the prognosis of affected patients. Based on our bioinformatics results, SOX2 overexpression is significantly associated with inferior disease-specific survival and worsened the progression-free interval of CRC patients. Our results demonstrate that zinc phthalocyanine (ZnPc)-PDT with 12 J/cm^2^ or 24 J/cm^2^ irradiation can substantially decrease tumor migration via downregulating MMP9 and ROCK1 and inhibit the clonogenicity of SW480 cells via downregulating CD44 and SOX2. Despite inhibiting clonogenicity, ZnPc-PDT with 12 J/cm^2^ irradiation fails to downregulate CD44 expression in SW480 cells. Our results indicate that ZnPc-PDT with 12 J/cm^2^ or 24 J/cm^2^ irradiation can substantially reduce the cell viability of SW480 cells and stimulate autophagy in the tumoral cells. Moreover, our results show that ZnPc-PDT with 12 J/cm^2^ or 24 J/cm^2^ irradiation can substantially arrest the cell cycle at the sub-G1 level, stimulate the intrinsic apoptosis pathway via upregulating caspase-3 and caspase-9 and downregulating Bcl-2. Indeed, our bioinformatics results show considerable interactions between the studied CSC-related genes with the studied migration- and apoptosis-related genes. Collectively, the current study highlights the potential role of ZnPc-PDT in inhibiting stemness and CRC development, which can ameliorate the prognosis of CRC patients.

## 1. Introduction

CRC is among the most commonly diagnosed malignancies in males and females worldwide [1]. Despite remarkable advances in treating CRC patients, the 5-year survival rate of CRC patients with metastatic and resistant tumors is lower than 5% [2]. Despite the emergence of chemo/radioresistance, the side effects of these approaches are not negligible [3,4]. Therefore, there is a need to develop a treatment for CRC patients with desirable response rates and favorable side effects.

PDT is a light-based approach that can be applied to the ablation of tumors. Following the application of a photosensitizer (PS) and visible light irradiation, the photodynamic reaction (PDR) can generate reactive oxygen species (ROS), which leads to tumor ablation [5,6,7]. One of the main advantages of this approach over conventional treatments is its minimal damage to non-tumoral cells. Indeed, PS tends to be concentrated in the tumoral cells; therefore, this approach does not lead to the side effects seen in conventional cancer therapies. Another advantage of this approach is its minimally invasive nature, which can be reflected in the reduced mortality rates of patients. Although the high efficacy and fewer side effects of this approach have paved the way for its translation into the clinics for treating various cancers, the resistance to PDT is still a challenge for treating CRC [8].

As a small population of tumoral cells, cancer stem cells (CSCs) have been implicated in developing therapy resistance [9,10]. Indeed, their self-renewal and stemness features are the main culprits in tumor recurrence and therapy resistance [11]. In addition, they can stimulate the epithelial-to-mesenchymal transition (EMT), which is associated with tumor migration [12,13]. Therefore, it is critical to identify and target CSCs to increase the response rate of PDT [14,15]. In this regard, metal phthalocyanines, e.g., zinc phthalocyanine (ZnPc), have been widely used as effective PS because of their high stability, low dark toxicity, high ROS production, and high optical toxicity capability [16,17,18]. Thus, it is critical to investigate the effect of ZnPc-PDT on the stemness feature of CRC cells and study its effect on CRC development. 

With a focus on CSCs, the current study aims to investigate the effect of ZnPc-PDT on the proliferation, apoptosis, autophagy, cell cycle, stemness, and migration of SW480 cells. Our results can pave the way for addressing the current daunting challenges in treating CRC.

## 2. Results

### 2.1. The Significance of CD44 and SOX2 in CRC Patients

Although there is a trend between CD44 overexpression and the poor disease-free interval of CRC patients, this trend is not statistically significant (*p* = 0.2909) (Figure 1A). Moreover, the elevated expression of SOX2 is significantly associated with the inferior disease-specific survival and progression-free interval of CRC patients (*p* = 0.004080, and *p* = 0.04581, respectively) (Figure 1B, and Figure 1C, respectively). Therefore, our in silico results indicate that the increased expression of SOX2, as a stem cell marker, is substantially associated with the poor prognosis of CRC patients.

### 2.2. The Effect of ZnPc-PDT on the Cell Viability of SW480 Cells

We studied the cytotoxicity effect of ZnPc-PDT on SW480 cells by measuring their cell viability after incubation with different concentrations of ZnPc (0.00017–8.651 μM) and light doses, i.e., 12 J/cm^2^ and 24 J/cm^2^. Our results show that administrating different doses of ZnPc and DMSO do not lead to the cytotoxic effect on SW480 cells compared to the control group (*p* > 0.05) (Figure 2A). In addition, 12 J/cm^2^ and 24 J/cm^2^ irradiations do not have any significant cytotoxic effects on SW480 cells compared to the control group (*p* > 0.05) (Figure 2B). However, the combination of ZnPC with irradiation significantly reduces the cell viability of SW480 cells (*p* < 0.0001) (Figure 2C,D). IC50 values of the ZnPc-PDT were calculated at 24 h after ZnPc-PDT by an MTT assay and a nonlinear regression. Moreover, the 24 J/cm^2^ irradiation is associated with a more pronounced cytotoxic effect than the dose of 12 J/cm^2^ following ZnPC treatment (Table 1).

### 2.3. The Effect of ZnPc-PDT on the Cell Cycle of SW480

Our results demonstrate that the ZnPc-PDT with 12 J/cm^2^ irradiation can significantly arrest the cell cycle of SW480 cells at the subG1 level compared to the control (Figure 3A) group (*p* < 0.0001) (Figure 3B,D). Our results show that ZnPc-PDT with 24 J/cm^2^ irradiation can also significantly arrest the cell cycle of SW480 cells at the subG1 level compared to the control (Figure 3A) group (*p* < 0.0001) (Figure 3C,D).

### 2.4. The Effect of ZnPc-PDT on the Autophagy of SW480 Cells

Our results show that ZnPc-PDT with 12 J/cm^2^ irradiation can significantly stimulate autophagy compared to the control (Figure 4A) group (*p* < 0.0001) (Figure 4B,D). In addition, ZnPc-PDT with 24 J/cm^2^ irradiation can significantly stimulate autophagy in the SW480 cells compared to the control (Figure 4A) group (*p* < 0.0001) (Figure 4C,D).

### 2.5. The Effect of ZnPc-PDT on Apoptosis/Necrosis

As shown in Figure 5A, ZnPc-PDT with 12 J/cm^2^ or 24 J/cm^2^ irradiation can significantly increase the apoptosis of SW480 cells compared to the control group (both *p* < 0.0001). In line with this, the results of DAPI staining also show increased cellular death following ZnPc-PDT with 12 J/cm^2^ or 24 J/cm^2^ irradiation (Figure 5B). Our results show that ZnPc-PDT with 12 J/cm^2^ irradiation can significantly upregulate the expression of caspase-9 compared to the control group (*p* < 0.001) (Figure 5C). Furthermore, ZnPc-PDT with 24 J/cm^2^ irradiation can significantly increase caspase-9 expression compared to the control group (*p* < 0.0001) (Figure 5C). Our results indicate that ZnPc-PDT with 12 J/cm^2^ or 24 J/cm^2^ irradiation can significantly upregulate the expression of caspase-3 and downregulate Bcl-2 compared to the control group (all *p* < 0.0001) (Figure 5C). However, these therapies do not significantly alter the expression of caspase-8 compared to the control group (both *p* > 0.05) (Figure 5C).

### 2.6. The Effect of ZnPc-PDT on the Stemness Features of SW480 Cells

We also studied the effect of ZnPc-PDT on the stemness features of SW480 cells via performing the colony-formation assay and investigating two crucial CSC markers, i.e., SOX2 and CD44. Our results show that ZnPc-PDT with 12 J/cm^2^ or 24 J/cm^2^ irradiation can substantially decrease the clonogenicity of SW480 cells (Figure 6A). Although ZnPc-PDT with 12 J/cm^2^ irradiation cannot significantly decrease CD44 expression compared to the control group, ZnPc-PDT with 24 J/cm^2^ irradiation significantly downregulate the expression of CD44 compared to the control group (*p* > 0.05, and *p* < 0.0001, respectively) (Figure 6B). Moreover, ZnPc-PDT with 12 J/cm^2^ or 24 J/cm^2^ irradiation can significantly downregulate SOX2 expression compared to the control group (both *p* < 0.0001) (Figure 6B).

### 2.7. The Effect of ZnPc-PDT on the Migration of SW480 Cells

As shown in Figure 7A, ZnPc-PDT with 12 J/cm^2^ or 24 J/cm^2^ irradiation can significantly downregulate MMP9 expression of SW480 cells compared to the control group (*p* < 0.001, and *p* < 0.0001, respectively). Moreover, our results indicate that ZnPc-PDT with 12 J/cm^2^ or 24 J/cm^2^ irradiation can significantly downregulate the ROCK1 expression of SW480 cells compared to the control group (both *p* < 0.0001) (Figure 7A). Consistent with these results, ZnPc-PDT with 12 J/cm^2^ or 24 J/cm^2^ irradiation substantially decreases the migration of SW480 cells compared to the control group (Figure 7B). 

### 2.8. The Interactions between the Studied Genes

Our results show a remarkable co-expression network between MMP9 and CD44, SOX2, and Bcl-2. Besides the remarkable gene interactions between caspase-8, caspase-9, caspase-3, and Bcl-2, these apoptotic-related genes have considerable interactions with the studied stemness-related genes, i.e., SOX2 and CD44, and the studied migration-related genes, i.e., MMP9 and ROCK1 (Figure 8).

## 3. Discussion

Although there have been remarkable advances in CRC treatment, tumor recurrence and metastasis are still daunting challenges. CSCs have been implicated in tumor recurrence, metastasis, cancer therapy resistance; therefore, targeting this unique population of tumor cells can pave the way for developing novel therapies for CRC patients. Our results indicate that ZnPc-PDT can substantially downregulate the expression of SOX2 and CD44, as crucial CSC markers, in SW480 cells. Besides inhibiting CSCs and clonogenicity of SW480 cells, ZnPc-PDT can remarkably decrease the migration of CRC cells, stimulate autophagy, arrest tumoral cells at the sub-G1 level, and activate the apoptosis of SW480 cells. Thus, ZnPc-PDT can substantially inhibit the stemness and development of CRC cells and might be a promising treatment approach for CRC. 

CD44 is a glycoprotein that is substantially upregulated in CSCs. The interaction between CD44 and its primary ligand, i.e., hyaluronic acid, has been implicated in tumor migration, proliferation, and tumor clonogenicity [10,12,19]. Cho et al. have reported that CD44 can activate the EMT and stimulate the PI3K/Akt pathway in SW480 cells [20]. Moreover, Park et al. have demonstrated that CD44 knockdown can downregulate the expression of Bcl-2 and upregulate the expression of cleaved caspase-3, caspase-8, and caspase-9 in colorectal cancer cells [21]. Wang et al. have shown that CD44 overexpression is significantly associated with lymph node metastasis and the inferior overall survival of CRC patients [22]. Although our results demonstrate that the CD44 overexpression is associated with inferior disease-free interval, this was not statistically significant. Of interest, we have found that ZnPc-PDT with 24 J/cm^2^ irradiation can substantially downregulate CD44 expression and inhibit the clonogenicity of CRC cells. 

SOX2, as a member of the SOX family, is an essential transcription factor that can maintain the pluripotency of stem cells in CRC [23]. Zheng et al. have shown that the knockdown of SOX2 can substantially inhibit the Rho-ROCK signaling pathway, resulting in reduced tumor invasion, inhibited clonogenicity, and a decreased stemness of CRC cells [24]. Han et al. have indicated that SOX2 silencing can inhibit the WNT pathway, and SOX2 overexpression can maintain the EMT process in CRC cells. They have shown that SOX2 overexpression is significantly associated with advanced tumor stage and liver metastasis [25]. Furthermore, Zhang et al. have indicated that SOX2 overexpression can lead to metastasis, resistance to apoptosis, chemoresistance, altered autophagy, and proliferation in tumoral cells [26]. A recent meta-analysis has shown that SOX2 overexpression is significantly associated with inferior overall survival, distant metastasis, poor differentiation, and advanced tumor stage in patients with CRC [27]. Consistent with these, our results have indicated that SOX2 overexpression is significantly associated with poor disease-specific survival and inferior progression-free interval. Of interest, our results demonstrate that ZnPc-PDT with 12 J/cm^2^ or 24 J/cm^2^ irradiation can substantially downregulate SOX2 expression and inhibit the clonogenicity of CRC cells. Moreover, ZnPc-PDT with 12 J/cm^2^ irradiation is associated with a more pronounced downregulation of SOX2 expression in CRC cells. 

As mentioned above, ZnPc is among the effective PS with low side effects for PDT. Doustvandi et al. have reported that ZnPc-PDT can substantially decrease the cell viability of tumoral cells [16]. In addition, it has been shown that ZnPc-PDT can remarkably stimulate the autophagy of tumoral cells and reduce their cell viability [17]. In CRC cells, Sekhejane et al. have reported that sulfonated ZnPC can predominantly localize in the lysosomes of tumoral cells and increase ROS generation, decreasing the cell viability of CRC cells [28]. In MCF-7 cells, ZnPc-PDT has also remarkably decreased the cell viability of tumoral cells and decreased tumor weight in mice bearing tumors [29]. In addition, ZnPc-PDT has considerably decreased the growth of A-431 cells and decreased tumor volume in animal models [30]. Consistent with these, our results show that ZnPc-PDT with 12 J/cm^2^ or 24 J/cm^2^ irradiation can result in a dose-dependent decrease in the cell viability of tumoral cells and stimulate autophagy in SW480 cells. 

Schmidt et al. have shown that ZnPc-PDT can lead to a dose-dependent decrease in tumor proliferation, cell cycle arrest at the sub-G1 level, and the stimulation of mitochondria-driven apoptosis pathway in tumoral cells. Indeed, there is a substantial upregulation of caspase-3 and remarkable downregulation of Bcl-2 following ZnPc-PDT in tumoral cells [31]. Xue et al. have demonstrated that nanoparticles loaded with ZnPc can considerably stimulate apoptosis in colorectal cancer cells. Furthermore, its intravenous administration has led to selective accumulation of ZnPc in the tumoral cells in mice bearing colorectal cancer cells, which is attributable to the conjugation of specific peptides in the utilized nanoparticles [32]. In osteosarcoma, Yu et al. have reported that ZnPc-PDT can remarkably downregulate Bcl-2 expression and arrest the cell cycle at the G2/M level [33]. In mice bearing Kyse-140, ZnPc-PDT has been associated with a decreased tumor volume in affected animal models. Moreover, ZnPc-PDT has increased the activity of caspase-3, increased apoptosis rate, and inhibited angiogenesis in Kyse-140 cells [34]. Consistent with these, our results indicate that ZnPc-PDT with 12 J/cm^2^ or 24 J/cm^2^ irradiation can substantially arrest the cell cycle at the sub-G1 level and stimulate the intrinsic apoptosis pathway via upregulating caspase-9 and caspase-3 and downregulating Bcl-2. Of interest, our bioinformatics results show that there are considerable interactions between these apoptosis-related genes with the studied CSC genes, i.e., CD44 and SOX2.

Recently, Yu et al. have reported that nanoparticles loaded with ZnPc with irradiation can substantially inhibit the migration of tumoral cells in vitro, and its intravenous administration can decrease tumor growth in mice bearing tumors [33]. Doustvandi et al. have shown that ZnPC-PDT can substantially decrease the migration of tumoral cells and downregulate the expression of vimentin and MMP9 in tumoral cells [17]. In SMMC-7721 and Huh7 cells, ZnPC-PDT has substantially decreased cell viability, inhibited clonogenicity, stimulated apoptosis, and decreased tumor weight in animal models. Furthermore, the fabrication of ZnPc/sorafenib in bovine serum albumin has remarkably decreased the migration of SMMC-7721 and Huh7 cells [35]. Consistent with these, our results demonstrate that ZnPc-PDT with 12 J/cm^2^ or 24 J/cm^2^ irradiation can inhibit the tumor migration and downregulate the expression of ROCK1 and MMP9 in SW480 cells. Of interest, our bioinformatics results indicate that there are remarkable interactions between the studied CSC-related genes, i.e., CD44 and SOX2, with these migration-related genes.

## 4. Materials and Methods

### 4.1. Materials

All materials were obtained from commercial sources and used without further purification. ZnPc was purchased from Sigma-Aldrich (CAS: 14320-04-8; St. Louis, MO, USA). Due to the hydrophobic nature of the ZnPc, dimethylsulfoxide (DMSO) was used to dissolve ZnPc in the RPMI1640 medium [36]. The Structure of ZnPc and UV–vis absorption spectrum of ZnPc in DMSO was measured by a NanoDrop 2000C spectrophotometer (Thermo Scientific, Waltham, MA, USA) at room temperature (Figure 9A,B). ZnPc has a weak absorption peak at 345 nm and a strong absorption peak at 675 nm. To match the light source wavelength with the maximum absorption of ZnPc, a continuous wave diode laser (Shenzhen Taiyong Technology, Shenzhen, China) with a wavelength of 675 nm was used. The cancer cells were imaged by an inverted microscope (Optika, Ponteranica, Italy) and a live cell imaging system (Citation 5, Biotek, CA, USA). Cell viability was analyzed by an ELISA reader (Sunrise ELISA Plate Reader, Tecan, Salzberg, Austria). Flow cytometric analysis was performed using a flow cytometer (MacsQuant Analyser 10, Miltenyi Biotech, Bergisch Gladbach, Germany). Quantitative real-time PCR (qRT-PCR) analysis was performed with a StepOnePlus real-time PCR system (Applied Biosystems, Foster City, CA, USA).

### 4.2. Cell Culture

SW480 cells, as human colorectal adenocarcinoma cells, were procured from the Pasteur Institute of Iran and these cells were cultured in RPMI1640 medium with 10% fetal bovine serum (FBS) (Gibco, Amarillo, TX, USA) and 1% penicillin/streptomycin (Gibco, USA). Then, CRC cells were maintained in the incubator at 37 °C with 5% CO_2_ and 95% humidity. The CRC cells were at the maximum of the logarithmic growth phase, and the number of passages was three for all assays.

### 4.3. PDT Treatment

#### 4.3.1. Photosensitizer

The first stock solution of ZnPc (8.651 μM) was prepared using DMSO and RPMI1640. Then, it was sonicated in a sonicator (bath type, Elma transonic T420, Singen am Hohentwiel, Germany). The final concentration of DMSO in the first stock solution was maintained at around 2% (*v/v*) in the RPMI1640 medium. Subsequently, the other concentrations were prepared by sequential diluting the first stock solution of ZnPc with the RPMI1640 medium.

#### 4.3.2. Light Source

A diode Laser (Shenzhen Taiyong Technology, China) with a wavelength of 675 nm and 80 mW output power and at an irradiance of 407 mW/cm^2^ was used to irradiate at different times, (30 and 60 s) the sample SW480 cells. The output power of the laser was measured by a power meter (PMC-121, ASHA, Iran). Parameters of the used laser are displayed in Table 2.

#### 4.3.3. PDT Treatment

For PDT treatment, the SW480 cells were classified into 5 different groups. The first group received neither ZnPc nor light source (control group). The second group was incubated only with different (decreasing) concentrations of DMSO (equal to the amount of DMSO in each concentration of ZnPc) without light source irradiation. The third group was incubated only with different concentrations of ZnPc (0.00017–8.651 μM) without light source irradiation. The fourth group received only light source irradiation (12 and 24 J/cm^2^). The final group, defined as the ZnPc-PDT group, was received both ZnPc (0.00017–8.651 μM) and light source irradiation (12 and 24 J/cm^2^).

In this manner, the SW480 cells were incubated by various concentrations of ZnPc (0.00017–8.651 μM) for 24h. Subsequently, the SW480 cells were washed twice with phosphate-buffered saline (PBS). Finally, the ZnPc absorbed by the SW480 cells were excited by a diode laser (675 nm) at two light doses (12 J/cm^2^ with duration of irradiation 30 s and 24 J/cm^2^ with duration of irradiation 60 s) in the dark and sterile conditions. The different tests were carried out 24 h after irradiation with the light source. In all experiments, ZnPc-PDT with 12 J/cm^2^ and ZnPc-PDT with 24 J/cm^2^ were separately performed and studied.

### 4.4. MTT Assay

In all experiments, SW480 cells were treated like that in the PDT treatment. The MTT assay was done to study the cytotoxicity effects of ZnPc in the absence of irradiation, ZnPc in the presence of irradiation, and light source alone on CRC cells. Briefly, the SW480 cells were seeded and incubated overnight. Following administration of two different doses of ZnPc-PDT, the viability of SW480 cells was determined by tetrazolium salt. In this method, 24 h after ZnPc-PDT, the SW480 cells were treated with an MTT solution (2 mg/mL), and after 4 h, 200 µL of DMSO was added to each well. Eventually, the optical density (OD) of each well was analyzed by an ELISA reader at 570 nm.

### 4.5. Flow Cytometry of Cell Cycle

After ZnPc-PDT, the cell cycle of SW480 cells was investigated by flow cytometry. Following ZnPc-PDT, a flow cytometry analysis of propidium-iodide (PI) staining was carried out; resuspended SW480 cells were fixed with 75% ethanol at 4 °C for 24 h. Then, the SW480 cells were treated with RNase A (Carl Roth, Karlsruhe, Germany) for 30 min at 37 °C. Finally, the SW480 cells were stained by a PI staining solution (0.01% Triton X-100 and 0.01% PI). After 10 min in a dark room at around 20–22 °C, the cell cycle of SW480 cells was investigated.

### 4.6. Flow Cytometry Analysis of Autophagic Cells

Autophagic vacuoles were detected using monodansylcadaverine as a fluorescent marker (MDC, Sigma Aldrich, Darmstadt, Germany). Following 24 h after ZnPc-PDT, the SW480 cells were incubated with 50 μM MDC at 37 °C for 10 min. Then, the MDC was removed, and SW480 cells were washed with PBS. Finally, autophagic cells were investigated immediately using a flow cytometer.

### 4.7. Apoptosis Assays

#### 4.7.1. Flow Cytometry Analysis for Necrotic/Apoptotic Cells

The percentage of necrotic/apoptotic cells was investigated by a flow cytometer. Following ZnPc-PDT, the SW480 cells were resuspended and stained by an ApoFlowEx^®^ FITC Kit (EXBIO, Vestec, Czech Republic). Then, 5 μL of annexin-V-FITC, 5 μL of PI, and 190 μL of binding buffer were added to each sample. After 10 min at room temperature in a dark room, the percentage of apoptotic/necrotic cells was studied by flow cytometry. 

#### 4.7.2. Nuclear Staining

The nuclear morphology changes were also studied using a live cell imaging system. After 24 h of administration of ZnPc-PDT, the SW480 cells were slowly washed with PBS. Then, tumoral cells were fixed with paraformaldehyde (4%) for 15 min. Next, these cells were washed thrice with PBS and treated with Triton X-100 (0.1%) for around 15 min. After another wash, SW480 cells were incubated with DAPI (0.1%) for 15 min in dark conditions. Subsequently, these cells were washed, and nuclear morphology changes were studied.

### 4.8. Colony Formation Assay

For this experiment, 1500 cells per well were seeded in 12-well plates. They were incubated to grow for several days in the incubator (37 °C, 5% CO_2_, and 95% humidity) until small colonies (about 50 cells per colony) could be seen. Then, the SW480 cells were treated and incubated for one week in the incubator. Subsequently, the SW480 colonies were incubated with crystal violet staining solution, and the number of colonies in each well was compared to the control group.

### 4.9. Migration Assay

The SW480 cells were scratched by a sterile yellow pipette tip to study the effect of ZnPc-PDT on their migration. At several times, i.e., 0 h, 12 h, and 24 h, the SW480 cells were photographed using an inverted microscope, and their migration was studied. 

### 4.10. Quantitative Real-Time PCR (qRT-PCR) Analysis

Twenty-four hours after treatments, the total RNA extraction was performed using RiboEx LS RNA (GeneAll Biotech, Seoul, Korea). Then, the cDNA of samples was obtained from RNAs using the cDNA synthesis kit (BioFact, Daejeon, Korea) to measure the expression of target mRNAs. Then, the qRT-PCR was performed with a standard SYBR Green PCR master mix (BioFact, Daejeon, Korea) protocol in the StepOnePlus real-time PCR. The GAPDH gene was used as a reference gene for SOX-2, CD44, ROCK1, MMP9, caspase-9, -8, -3, and Bcl-2 genes. The primer sequences used in this study are shown in Table 3. The relative expression levels of target genes were normalized to internal controls using the 2^−ΔΔCt^ cycle threshold method.

### 4.11. In Silico Study

To better understand the impact of CD44 and SOX2 on clinical features of CRC patients, their expression levels in primary tumors, normal tissues, metastatic tumors, and recurrent tumors were extracted from the TCGA using the UCSC Cancer Browser (https://xenabrowser.net/; accessed on 14 May 2021). Furthermore, the prognostic values of CD44 and SOX2 in patients with CRC were analyzed and demonstrated in Kaplan–Meier survival curves. Moreover, a co-expression network, consisting of the studied genes, and their co-expressed genes were constructed by GeneMANIA (https://genemania.org; accessed on 14 May 2021) to demonstrate the gene interactions.

### 4.12. Statistical Analysis

In this study, all assays were performed in 3 independent experiments. All results were presented as the means ± standard deviation (SD) and analyzed using GraphPad Prism software (Version 6, San Diego, CA, USA). To determine the statistically significant differences between the means of groups, *t*-test and one-way ANOVA were performed. *p* values < 0.05 were considered statistically significant.

## 5. Conclusions

CSCs are the main culprits for tumor recurrence, metastasis, and cancer therapy resistance; thus, there is a pressing need to target this unique population of tumoral cells. Our results indicate that the elevated expression of SOX2, as a cancer stem cell marker, is remarkably associated with the inferior disease-specific survival and a worsened progression-free interval of CRC patients, which can be used to determine the prognosis of CRC patients in the pathology department as a routine practice. Our study indicated remarkable interactions between these CSC-related genes with the studied apoptosis-related genes, i.e., caspase-9, caspase-3, and Bcl-2, and studied migration-related genes, i.e., MMP9 and ROCK1. The current study shows that ZnPc-PDT with 12 J/cm^2^ or 24 J/cm^2^ irradiation can substantially decrease the cell viability of tumoral cells and stimulate autophagy in SW480 cells. In addition, our results show that ZnPc-PDT with 12 J/cm^2^ or 24 J/cm^2^ irradiation can remarkably arrest the cell cycle at the sub-G1 level, stimulate the intrinsic apoptosis pathway via upregulating caspase-3 and caspase-9 and downregulating Bcl-2, inhibit the migration of tumoral cells via downregulating ROCK1 and MMP9, and decrease the stemness and clonogenicity of SW480 cells via downregulating CD44 and SOX2. Despite inhibiting clonogenicity, ZnPc-PDT with 12 J/cm^2^ irradiation has failed to decrease CD44 expression in SW480 cells. Overall, this approach might address the current challenges in treating CRC. 

## Figures and Tables

**Figure 1 molecules-26-06877-f001:**
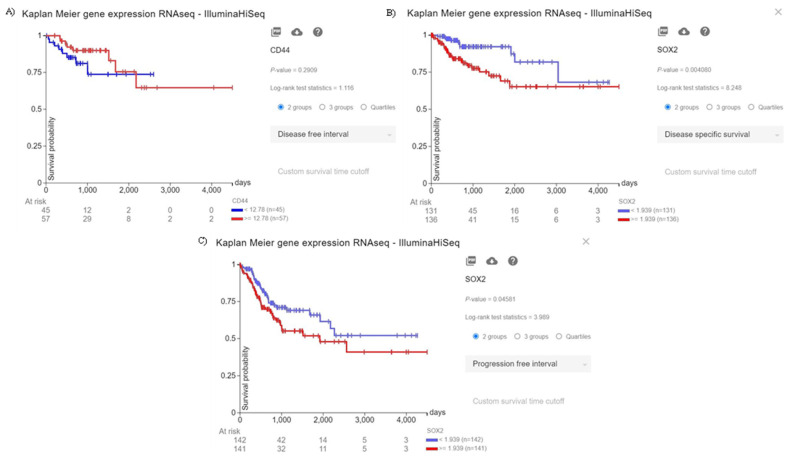
The prognostic value of CD44 and SOX2. (**A**) Although there is a trend between the increased expression of CD44 and the inferior disease-free interval of CRC patients, this trend is not statistically significant (*p* = 0.2909). (**B**) The increased expression of SOX2 is significantly associated with the poor disease-specific survival of CRC patients (*p* = 0.004080). (**C**) The increased expression of SOX2 is significantly associated with worsened progression-free interval of CRC patients (*p* = 0.04581).

**Figure 2 molecules-26-06877-f002:**
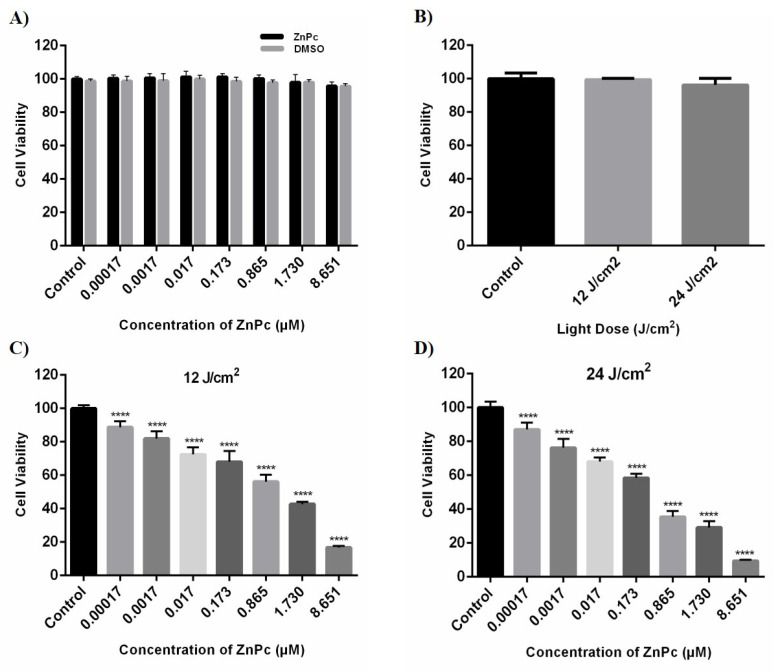
The cytotoxic effects of ZnPc, DMSO, light dose, and ZnPc-PDT were determined at 24 h after treatments by MTT assay and nonlinear regression on SW480 cells. (**A**) The percentage of cell viability in the presence of different concentrations of ZnPc and DMSO (without irradiation). (**B**) The percentage of cell viability after different irradiations of light dose (without ZnPc; 12 and 24 J/cm^2^ irradiation). (**C**) The percentage of cell viability after ZnPc-PDT with 12 J/cm^2^ irradiation. (**D**) The percentage of cell viability following ZnPc-PDT with 24 J/cm^2^ irradiation. The results are expressed as mean ± SD (*n* = 3); **** *p* < 0.0001 versus control.

**Figure 3 molecules-26-06877-f003:**
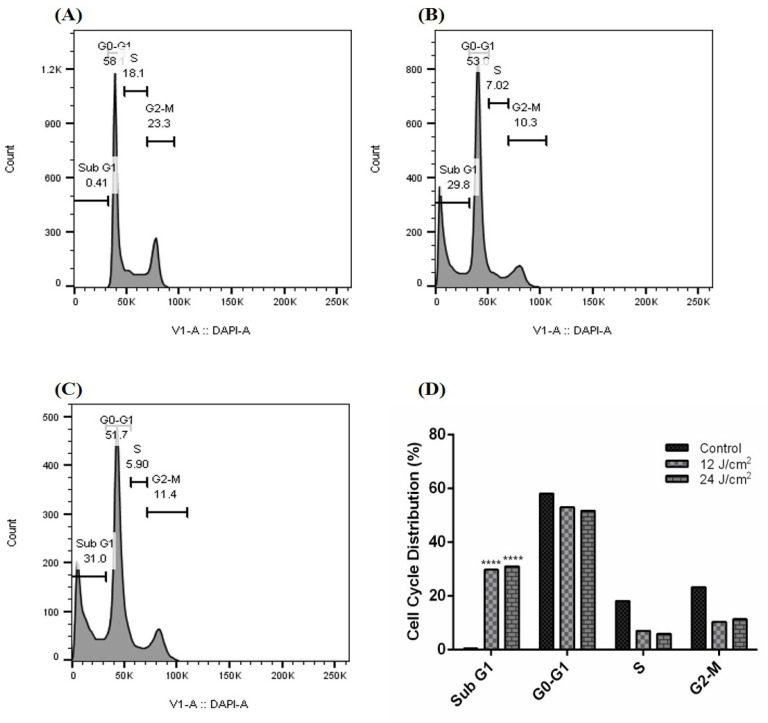
Cell cycle following ZnPc-PDT. (**A**) control cells; (**B**) cells were treated with ZnPc-PDT with 12 J/cm^2^ irradiation; (**C**) cells were treated with ZnPc-PDT with 24 J/cm^2^ irradiation; (**D**) the results are represented as mean ± SD (*n* = 3); **** *p* < 0.0001 compared with control cells.

**Figure 4 molecules-26-06877-f004:**
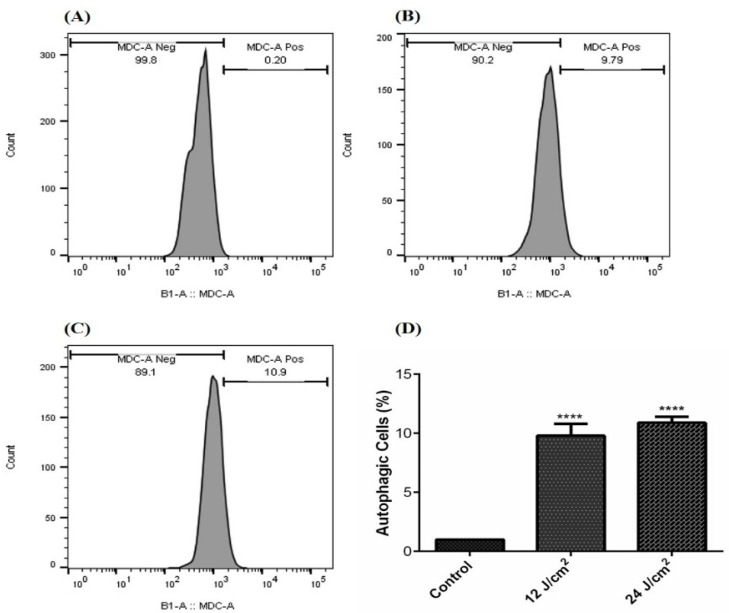
The percentage of autophagic cells following ZnPc-PDT; (**A**) control cells; (**B**) cells were treated with 12 J/cm^2^; (**C**) cells were treated with 24 J/cm^2^; (**D**) the results are expressed as mean ± SD (*n* = 3); **** *p* < 0.0001 compared with control group.

**Figure 5 molecules-26-06877-f005:**
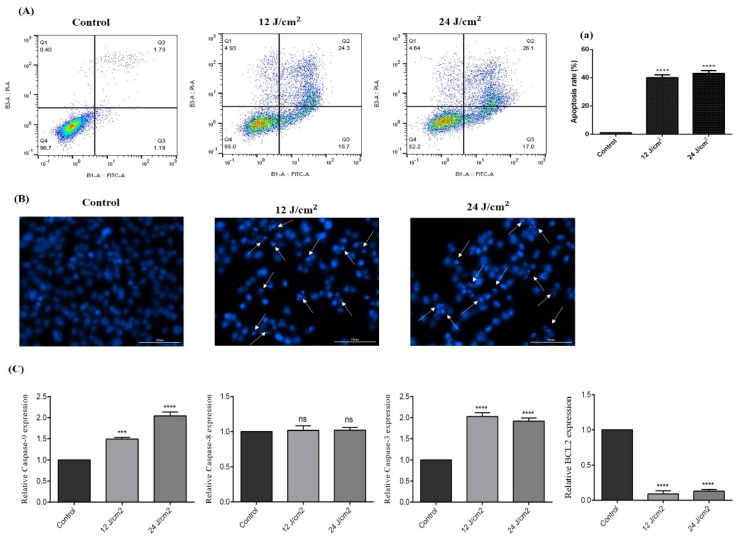
The percentage of apoptotic cells following ZnPc-PDT. Apoptosis was studied after ZnPc-PDT using (**A**,**a**) flow cytometry; (**B**) DAPI staining; (**C**) qRT-PCR. The results are expressed as mean ± SD (*n* = 3); *** *p* < 0.001, **** *p* < 0.0001 versus control.

**Figure 6 molecules-26-06877-f006:**
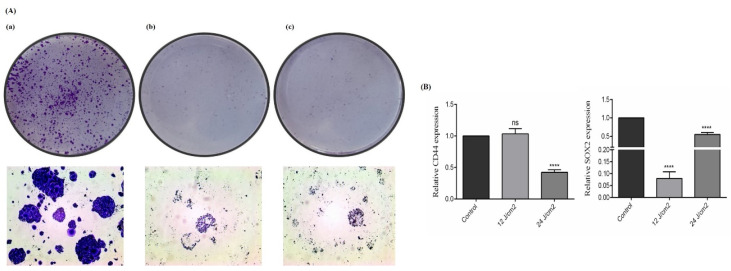
The stemness features of SW480 cells were evaluated after ZnPc-PDT. (**A**) The clonogenicity of tumoral cells was studied 24 h after ZnPc-PDT; (**a**) control cells; (**b**) cells were treated with 12 J/cm^2^ and (**c**) cells were treated with 24 J/cm^2^; (**B**) the effects of ZnPc-PDT on the expression levels of CD44 and SOX2 were studied in SW480 cells at 24 h after ZnPc-PDT; results are means ± SD of three independent investigations and **** *p* < 0.0001 versus control.

**Figure 7 molecules-26-06877-f007:**
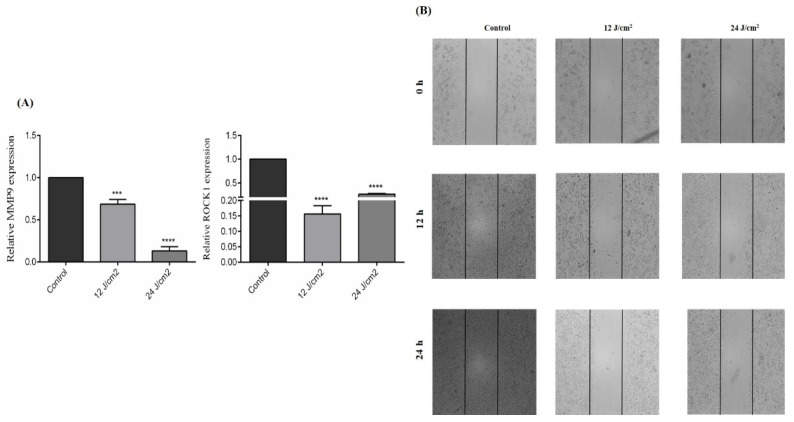
The effect of ZnPc-PDT on the migration of SW480 cells. (**A**) The effect of ZnPc-PDT on MMP9 and ROCK1 gene expression. Results are presented as means ± SD of 3 independent tests and, *** *p* < 0.001, **** *p* < 0.0001 versus control. (**B**) After ZnPc-PDT, wound gaps were studied at 0 h, 12 h, and 24 h.

**Figure 8 molecules-26-06877-f008:**
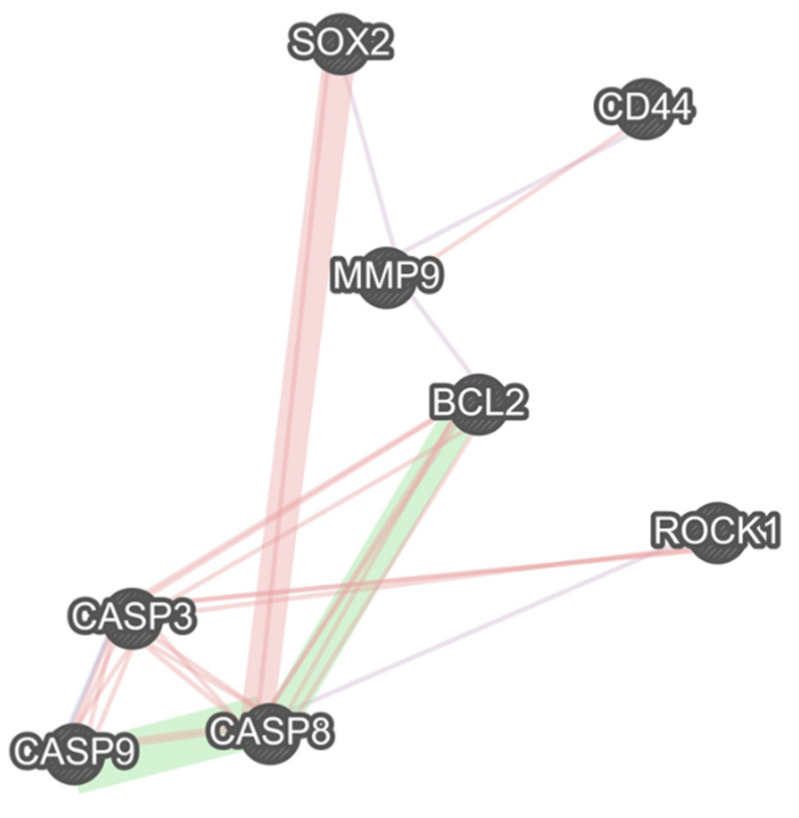
There are multiple remarkable associations between studied stemness-related, migration-related, and apoptosis-related genes. The purple links show the gene co-expression, the pink ones illustrate the physical interactions between the genes, and the green links show the genetic interactions.

**Figure 9 molecules-26-06877-f009:**
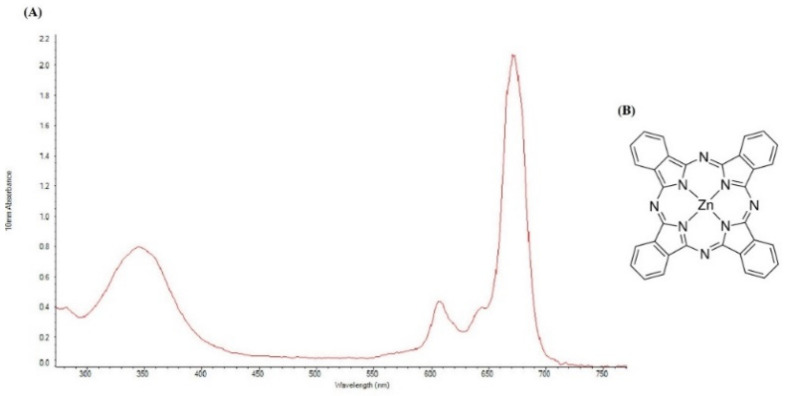
(**A**) UV–vis spectrum of 8.651 μM of ZnPc in DMSO. (**B**) Structure of ZnPc.

**Table 1 molecules-26-06877-t001:** IC50 values of ZnPc-PDT in different doses of the light dose.

Light Dose (J/cm^2^)	IC50 Values (μM)
12 J/cm^2^	0.628 ± 0.01
24 J/cm^2^	0.145 ± 0.01

**Table 2 molecules-26-06877-t002:** The parameters of laser used in this study.

Parameter	Continuous Wave Laser
Wavelength	675 (nm)
Wave emission	Continuous
Beam diameter	5 (mm)
Distance between laser tip and sample	151 (mm)
Output power	80 (mW)
Duration of irradiation	30, 60 (s)

**Table 3 molecules-26-06877-t003:** Primers sequences.

Primer Name	Primer Sequence
*GAPDH*	Forward: 5′ CCTCGTCCCGTAGACAAAA 3′
	Reverse: 5′ AATCTCCACTTTGCCACTG 3′
*Bcl2*	Forward: 5′ CCTGTGGATGACTGAGTACC 3′
	Reverse: 5′ GAGACAGCCAGGAGAAATCA 3′
*Caspase-9*	Forward: 5′ CCGGAATCCTGCTTGGGTATC 3′
	Reverse: 5′ CATCGGTGCATTTGGCATGTA 3′
*Caspase-8*	Forward: 5′ GGTCTGAAGGCTGGTTGTTC 3′
	Reverse: 5′ AATCTCAATATTCCCAAGGTTCAAG 3′
*Caspase-3*	Forward: 5′ TGTCATCTCGCTCTGGTACG 3′
	Reverse: 5′ AAATGACCCCTTCATCACCA 3′
*MMP9*	Forward: 5′ GGTTCTTCTGCGCTACTGCTG 3′
	Reverse: 5′ GTCGTAGGGCTGCTGGAAGG 3′
*ROCK1*	Forward: 5′ AATCGTGTGGGATGCTACCT 3′
	Reverse: 5′ AAAACCCTCAGTGTGTTGTGC 3′
*CD44*	Forward: 5′ CTGCCGCTTTGCAGGTGTA 3′
	Reverse: 5′ CATTGTGGGCAAGGTGCTATT 3′
*SOX2*	Forward: 5′ACATGTGAGGGCCGGACAGC 3′
	Reverse: 5′TTGCGTGAGTGTGGATGGGATTGG 3′

## Data Availability

Not applicable.

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
