# Peer review of "Photodynamic Therapy with Zinc Phthalocyanine Inhibits the Stemness and Development of Colorectal Cancer: Time to Overcome the Challenging Barriers?"

_molecules, 2021, doi:10.3390/molecules26226877_

Round 1
Reviewer 1 Report
Gholizadeh et al have shown that targeted Zinc phthalocyanine-assisted photodynamic therapy can be beneficial for colon-rectal cancer patients. The design of the work is convincing. And the results are promising. However, there are no characterizations of Zinc phthalocyanine sizes, shapes conjugations, and stability. In addition, I didn’t find any information on the duration of the photodynamic irradiation.
Minor point, the quality of the figures needed to be improved and enlarged.
Author Response
The authors appreciate your kind attention.

Reviewer 2 Report
In this study the authors report a complete study based on the Photodynamic Therapy (PDT) of colorectal cancer using Zinc-phthalocyanine (ZnPC) as photosensitizer. This is an interesting and very complete study with remarkable findings. The authors showed the therapeutic approach based on ZnPC-PDT reduce the cell viability of SW480 cells, stimulate the autophagy in the tumoral cells and decrease the tumor migration. Moreover, the authors suggested that this approach inhibit the clonogenicity of SW480 cell line and stimulate the intrinsic apoptosis. However, some points must be clarified in order to improve the manuscript quality.
Since one of the major requisites of PDT is the selectivity of the photosensitizer for the tumoral cells and that is expected that the normal cells do not suffer any photodynamic effect, how authors justify that the ZnPC could be applied as photodynamic therapeutic agent? There is any evidence that this photosensitizer is selective for the tumoral cell line (SW482)?
Authors does not refer also if ZnPC induces dark toxicity? There is any evidence that this photosensitizer is not toxic in the absence of light?
In the PDT treatment of SW480 cells with ZnPC study, there is no information the about the origin (it is commercial?) and preparation of stock solution of the ZnPC (which was the solvent?). It would be interesting if the authors could provide the exact procedure of the incubation of cells with ZnPC a figure that shows the overlap between the absorption spectra of ZnPC and the spectral irradiance of the light source used.
Author Response

(The authors gave the same response as above.)

Reviewer 3 Report
This report relates to the use of photodynamic therapy (PDT) for treatment of colorectal cancer. Effects of PDT include both direct killing and a shutdown of the tumor vasculature. There can also be an enhanced immune recognition of malignant cell types as a byproduct of treatment. PDT-induced cell death usually results from photodamage to sub-cellular sites resulting in the evoking of programmed death pathways. This report only deals with direct photokilling. One factor in PDT is the need for light, with optical properties of tissues posing a problem to light transmission. Some ambiguities need to be cleared up before this is appropriate for publication.
The agent chosen is not in current clinical use. If this is being proposed as potentially useful for clinical study, it might be useful to eventually compare its properties with agents that already have regulatory approval for human use. From the data, as explained below, it is not clear whether the authors imply that a sub-population of stem cells is present or whether the results shown apply to the entire cell population. Since some of the proposed markers are affected and others are not, it is not clear what message this report conveys. If a cell population is eradicated, it will not matter whether the dying cells express a protein. If the population is not wholly eradicated, it is also not clear whether further prognosis will be related to the presence of marker proteins.
Figure 1 is difficult to read since some of the fonts used are quite small and the contrast is low. The legend does not indicate what is being examined. This needs to be modified before the relevance of these data can be assessed. What is ‘irritation’ (line 96)? The legend to Fig. 2 should mention how ‘viability’ is assessed. It appears that light and sensitizer alone have no effect but that photodynamic lethality can be demonstrated and that efficacy increases with the light or drug dose. It might be more useful to see log dose-response curves since these can reveal whether there is a shoulder on the curve, consistent with cytoprotection by autophagy.
In Table 2, it is the light dose that is relevant, not the ‘laser’ dose. The laser is a light-producing device. Any other light-emitting device would work as well. This applies throughout the manuscript. The legend to Fig. 2 does not indicate at what time interval these measurements were acquired. Fig. 5 tends to have numbers too small to be read. This is also true for the cell images. It is barely possible to make out apoptotic morphology of the nuclei. It might be more informative to report DEVDase activity. Fig. 6 shows a significant loss of clonogenicity after PDT. It is not indicated at what time interval CD44 and SOX2 levels were determined. If there are substantial numbers of dead and dying cells, it is not clear what ‘relative expression’ means. Do these values reflect protein levels in surviving cells?
It is important not to confuse MTT assay results with clonogenicity measurements. The MTT assay measures the activity of some mitochondrial dehydrogenases. This can often be correlated with survival but needs to be expressed in terms of cytotoxicity, not survival, since survival is not being measured. Depending on the time at which the assay is carried out, there may be delayed death or recovery. Clonogenicity is unambiguous. It is well-known that PDT can kill cells, that apoptosis is a common factor, that autophagy is often evoked as a cytoprotective process unless lysosomes are targeted for photodamage. At what time after irradiation was the PCR analysis carried out?
Autophagy is monitored by numbers of MDC-labeled cells. This is among the older procedures, and in the comprehensive ‘Guidelines’ article published in the journal Autophagy, the limitations of this procedure are described (Autophagy 8:4, 445–544; April 2012). This is only a minor aspect of this report and is not a major concern. It s likely that there will be some autophagic response to PDT.
Author Response

(The authors gave the same response as above.)
